# Deep Convolutional Networks as shallow Gaussian Processes

**Adrià Garriga-Alonso**
department of Engineering
University of Cambridge
ag919@cam.ac.uk

**Carl Edward Rasmussen**
Department of Engineering
University of Cambridge
cer54@cam.ac.uk

**Laurence Aitchison**
Department of Engineering
University of Cambridge
laurence.aitchison@gmail.com

## Abstract

We show that the output of a (residual) convolutional neural network (CNN) with an appropriate prior over the weights and biases is a Gaussian process (GP) in the limit of infinitely many convolutional filters, extending similar results for dense networks. For a CNN, the equivalent kernel can be computed exactly and, unlike "deep kernels", has very few parameters: only the hyperparameters of the original CNN. Further, we show that this kernel has two properties that allow it to be computed efficiently; the cost of evaluating the kernel for a pair of images is similar to a single forward pass through the original CNN with only one filter per layer. The kernel equivalent to a 32-layer ResNet obtains 0.84% classification error on MNIST, a new record for GPs with a comparable number of parameters. [1]

## 1 Introduction

Convolutional Neural Networks (CNNs) have powerful pattern-recognition capabilities that have recently given dramatic improvements in important tasks such as image classification (Krizhevsky et al., 2012). However, as CNNs are increasingly being applied in real-world, safety-critical domains, their vulnerability to adversarial examples (Szegedy et al., 2013; Kurakin et al., 2016), and their poor uncertainty estimates are becoming increasingly problematic. Bayesian inference is a theoretically principled and demonstrably successful (Snoek et al., 2012; Deisenroth & Rasmussen, 2011) framework for learning in the face of uncertainty, which may also help to address the problems of adversarial examples (Gal & Smith, 2018). Unfortunately, Bayesian inference in CNNs is extremely difficult due to the very large number of parameters, requiring highly approximate factorised variational approximations (Blundell et al., 2015; Gal & Ghahramani, 2015), or requiring the storage (Lakshminarayanan et al., 2017) of large numbers of posterior samples (Welling & Teh, 2011; Mandt et al., 2017).

Other methods such as those based on Gaussian Processes (GPs) are more amenable to Bayesian inference, allowing us to compute the posterior uncertainty exactly (Rasmussen & Williams, 2006). This raises the question of whether it might be possible to combine the pattern-recognition capabilities of CNNs with exact probabilistic computations in GPs. Two such approaches exist in the literature. First, deep convolutional kernels (Wilson et al., 2016) parameterise a GP kernel using the weights and biases of a CNN, which is used to embed the input images into some latent space before computing their similarity. The CNN parameters of the resulting kernel then have to be optimised by gradient descent. However, the large number of kernel parameters in the CNN reintroduces the risk of overconfidence and overfitting. To avoid this risk, we need to infer a posterior over the CNN kernel parameters, which is as difficult as directly inferring a posterior over the parameters of the original CNN. Second, it is possible to define a convolutional GP (van der Wilk et al., 2017) or a

---

[1]Code to replicate this paper is available at https://github.com/convnets-as-gps/convnets-as-gps

deep convolutional GP (Kumar et al., 2018) by defining a GP that takes an image patch as input, and using that GP as a component in a larger CNN-like system. However, inference in such systems is very computationally expensive, at least without the use of potentially severe variational approximations (van der Wilk et al., 2017).

An alternative approach is suggested by the underlying connection between Bayesian neural networks (NNs) and GPs. In particular, Neal (1996) showed that the function defined by a single-layer fully-connected NN with infinitely many hidden units, and random independent zero-mean weights and biases is equivalent to a GP, implying that we can do exact Bayesian inference in such a NN by working with the equivalent GP. Recently, this result was extended to arbitrarily deep fully-connected NNs with infinitely many hidden units at each layer (Lee et al., 2017; Matthews et al., 2018a). However, these fully-connected networks are rarely used in practice, as they are unable to exploit important properties of images such as translational invariance, raising the question of whether state-of-the-art architectures such as CNNs (LeCun et al., 1990) and ResNets (He et al., 2016a) have equivalent GP representations. Here, we answer in the affirmative, giving the GP kernel corresponding to arbitrarily deep CNNs and to (convolutional) residual neural networks (He et al., 2016a). In this case, if each hidden layer has an infinite number of convolutional *filters*, the network prior is equivalent to a GP.

Furthermore, we show that two properties of the GP kernel induced by a CNN allow it to be computed very efficiently. First, in previous work it was necessary to compute the covariance matrix for the output of a single convolutional filter applied at all possible locations within a single image (van der Wilk et al., 2017), which was prohibitively computationally expensive. In contrast, under our prior, the downstream weights are independent with zero-mean, which decorrelates the contribution from each location, and implies that it is necessary only to track the patch variances, and not their covariances. Second, while it is still necessary to compute the variance of the output of a convolutional filter applied at all locations within the image, the specific structure of the kernel induced by the CNN means that the variance at every location can be computed simultaneously and efficiently as a convolution.

Finally, we empirically demonstrate the performance increase coming from adding translation-invariant structure to the GP prior. Without computing any gradients, and without augmenting the training set (e.g. using translations), we obtain 0.84% error rate on the MNIST classification benchmark, setting a new record for nonparametric GP-based methods.

## 2 GP BEHAVIOUR IN A CNN

For clarity of exposition, we will treat the case of a 2D convolutional NN. The result applies straightforwardly to $n$D convolutions, dilated convolutions and upconvolutions ("deconvolutions"), since they can be represented as linear transformations with tied coefficients (see Fig. 1).

### 2.1 A 2D CONVOLUTIONAL NETWORK PRIOR

The network takes an arbitrary input image $\mathbf{X}$ of height $H^{(0)}$ and width $D^{(0)}$, as a $C^{(0)} \times (H^{(0)} D^{(0)})$ real matrix. Each row, which we denote $\mathbf{x}_1, \mathbf{x}_2, \ldots, \mathbf{x}_{C^{(0)}}$, corresponds to a channel of the image (e.g. $C^{(0)} = 3$ for RGB), flattened to form a vector. The first activations $\mathbf{A}^{(1)}(\mathbf{X})$ are a linear transformation of the inputs. For $i \in \{1, \ldots, C^{(1)}\}$:

$$\mathbf{a}_i^{(1)}(\mathbf{X}) := b_i^{(1)} \mathbf{1} + \sum_{j=1}^{C^{(0)}} \mathbf{W}_{i,j}^{(1)} \mathbf{x}_j .$$ (1)

We consider a network with $L$ hidden layers. The other activations of the network, from $\mathbf{A}^{(2)}(\mathbf{X})$ up to $\mathbf{A}^{(L+1)}(\mathbf{X})$, are defined recursively:

$$\mathbf{a}_i^{(\ell+1)}(\mathbf{X}) := b_i^{(\ell+1)} \mathbf{1} + \sum_{j=1}^{C^{(\ell)}} \mathbf{W}_{i,j}^{(\ell+1)} \phi \left( \mathbf{a}_j^{(\ell)}(\mathbf{X}) \right) .$$ (2)

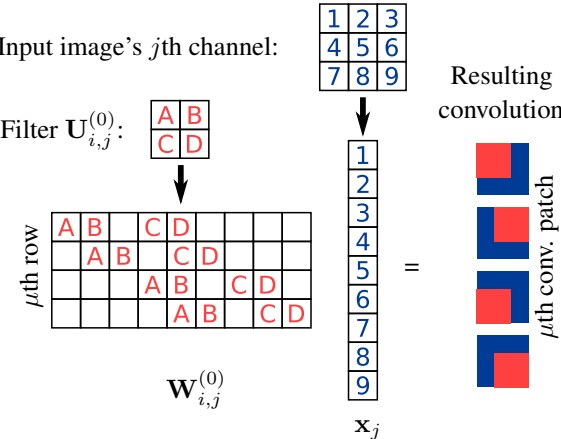

Figure 1: The 2D convolution $\mathbf{U}_{i,j}^{(0)} * \mathbf{x}_j$ as the dot product $\mathbf{W}_{i,j}^{(0)}\mathbf{x}_j$. The blank elements of $\mathbf{W}_{i,j}^{(0)}$ are zeros. The $\mu$th row of $\mathbf{W}_{i,j}^{(0)}$ corresponds to applying the filter to the $\mu$th convolutional patch of the channel $\mathbf{x}_j$.

The activations $\mathbf{A}^{(\ell)}(\mathbf{X})$ are $C^{(\ell)} \times (H^{(\ell)}D^{(\ell)})$ matrices. Each row $\mathbf{a}_i^{(\ell+1)}$ represents the flattened $j$th channel of the image that results from applying a convolutional filter to $\phi(\mathbf{A}^{(\ell)}(\mathbf{X}))$.

The structure of the pseudo-weight matrices $\mathbf{W}_{i,j}^{(\ell+1)}$ and biases $b_i^{(\ell+1)}$, for $i \in \{1, \ldots, C^{(\ell+1)}\}$ and $j \in \{1, \ldots, C^{(\ell)}\}$, depends on the architecture. For a convolutional layer, each row of $\mathbf{W}_{i,j}^{(\ell+1)}$ represents a *position* of the filter, such that the dot product of all the rows with the image vector $\mathbf{x}_j$ represents applying the convolutional filter $\mathbf{U}_{i,j}^{(\ell+1)}$ to the $j$th channel. Thus, the elements of each row of $\mathbf{W}_{i,j}^{(\ell+1)}$ are: 0 where the filter does not apply and the corresponding element of $\mathbf{U}_{i,j}^{(\ell+1)}$ where it does, as illustrated in Fig. 1.

The outputs of the network are the last activations, $\mathbf{A}^{(L+1)}(\mathbf{X})$. In the classification or regression setting, the outputs are not spatially extended, so we have $H^{(L+1)} = D^{(L+1)} = 1$, which is equivalent to a fully-connected output layer. In this case, the pseudo-weights $\mathbf{W}_{i,j}^{(L+1)}$ only have one row, and the activations $\mathbf{a}_i^{(L+1)}$ are single-element vectors.

Finally, we define the prior distribution over functions by making the filters $\mathbf{U}_{i,j}^{(\ell)}$ and biases $b_i^{(\ell)}$ be independent Gaussian random variables (RVs). For each layer $\ell$, channels $i, j$ and locations within the filter $x, y$:

$$U_{i,j,x,y}^{(\ell)} \sim \mathcal{N}\left(0, \sigma_w^2/C^{(\ell)}\right), \qquad\qquad b_i^{(\ell)} \sim \mathcal{N}\left(0, \sigma_b^2\right). \qquad (3)$$

Note that, to keep the activation variance constant, the weight variance is divided by the number of input channels. The weight variance can also be divided by the number of elements of the filter, which makes it equivalent to the NN weight initialisation scheme introduced by He et al. (2016a).

## 2.2 Argument for GP behaviour

We follow the proofs by Lee et al. (2017) and Matthews et al. (2018a) to show that the output of the CNN described in the previous section, $\mathbf{A}^{(L+1)}$, defines a GP indexed by the inputs, $\mathbf{X}$. Their proof (Lee et al., 2017) proceeds by applying the multivariate Central Limit Theorem (CLT) to each layer in sequence, i.e. taking the limit as $N^{(1)} \to \infty$, then $N^{(2)} \to \infty$ etc, where $N^{(\ell)}$ is the number of hidden units in layer $\ell$. By analogy, we sequentially apply the multivariate CLT by taking the limit as the number of channels goes to infinity, i.e. $C^{(1)} \to \infty$, then $C^{(2)} \to \infty$ etc. While this is the simplest approach to taking the limits, other potentially more realistic approaches also exist (Matthews et al., 2018a).

The fundamental quantity we consider is a vector formed by concatenating the feature maps (or equivalently channels), $\mathbf{a}_j^{(\ell)}(\mathbf{X})$ and $\mathbf{a}_j^{(\ell)}(\mathbf{X}')$ from data points $\mathbf{X}$ and $\mathbf{X}'$,

$$\mathbf{a}_i^{(\ell)}(\mathbf{X}, \mathbf{X}') = \begin{pmatrix} \mathbf{a}_i^{(\ell)}(\mathbf{X}) \\ \mathbf{a}_i^{(\ell)}(\mathbf{X}') \end{pmatrix}. \tag{4}$$

This quantity (and the following arguments) can all be extended to the case of countably finite numbers of input points.

**Induction base case.** For any pair of data points, $\mathbf{X}$ and $\mathbf{X}'$ the feature-maps corresponding to the $j$th channel, $\mathbf{a}_j^{(1)}(\mathbf{X}, \mathbf{X}')$ have a multivariate Gaussian joint distribution. This is because each element is a linear combination of shared Gaussian random variables: the biases, $\mathbf{b}_j^{(0)}$ and the filters, $\mathbf{U}_{j,:}^{(0)}$. Following Eq. (1),

$$\mathbf{a}_i^{(1)}(\mathbf{X}, \mathbf{X}') = b_i^{(1)}\mathbf{1} + \sum_{i=1}^{C^{(0)}} \begin{pmatrix} \mathbf{W}_{i,j}^{(1)} & \mathbf{0} \\ \mathbf{0} & \mathbf{W}_{i,j}^{(1)} \end{pmatrix} \begin{pmatrix} \mathbf{x}_i \\ \mathbf{x}_i' \end{pmatrix}, \tag{5}$$

where $\mathbf{1}$ is a vector of all-ones. While the elements within a feature map display strong correlations, different feature maps are independent and identically distributed (iid) conditioned on the data (i.e. $\mathbf{a}_i^{(1)}(\mathbf{X}, \mathbf{X}')$ and $\mathbf{a}_{i'}^{(1)}(\mathbf{X}, \mathbf{X}')$ are iid for $i \neq i'$), because the parameters for different feature-maps (i.e. the biases, $b_i^{(1)}$ and the filters, $\mathbf{W}_{i,:}^{(1)}$) are themselves iid.

**Induction step.** Consider the feature maps at the $\ell$th layer, $\mathbf{a}_j^{(\ell)}(\mathbf{X}, \mathbf{X}')$, to be iid multivariate Gaussian RVs (i.e. for $j \neq j'$, $\mathbf{a}_j^{(\ell)}(\mathbf{X}, \mathbf{X}')$ and $\mathbf{a}_{j'}^{(\ell)}(\mathbf{X}, \mathbf{X}')$ are iid). Our goal is to show that, taking the number of channels at layer $\ell$ to infinity (i.e. $C^{(\ell)} \to \infty$), the same properties hold at the next layer (i.e. all feature maps, $\mathbf{a}_i^{(\ell+1)}(\mathbf{X}, \mathbf{X}')$, are iid multivariate Gaussian RVs). Writing eq. (2) for two training examples, $\mathbf{X}$ and $\mathbf{X}'$, we obtain,

$$\mathbf{a}_i^{(\ell+1)}(\mathbf{X}, \mathbf{X}') = b_i^{(\ell+1)}\mathbf{1} + \sum_{j=1}^{C^{(\ell)}} \begin{pmatrix} \mathbf{W}_{i,j}^{(\ell+1)} & \mathbf{0} \\ \mathbf{0} & \mathbf{W}_{i,j}^{(\ell+1)} \end{pmatrix} \phi(\mathbf{a}_j^{(\ell)}(\mathbf{X}, \mathbf{X}')) \tag{6}$$

We begin by showing that $\mathbf{a}_i^{(\ell+1)}(\mathbf{X}, \mathbf{X}')$ is a multivariate Gaussian RV. The first term is multivariate Gaussian, as it is a linear function of $b_i^{(\ell+1)}$, which is itself iid Gaussian. We can apply the multivariate CLT to show that the second term is also Gaussian, because, in the limit as $C^{(\ell)} \to \infty$, it is the sum of infinitely many iid terms: $\mathbf{a}_j^{(\ell)}(\mathbf{X}, \mathbf{X}')$ are iid by assumption, and $\mathbf{W}_{i,j}^{(\ell+1)}$ are iid by definition. Note that the same argument applies to all feature maps jointly, so all elements of $\mathbf{A}^{(\ell+1)}(\mathbf{X}, \mathbf{X}')$ (defined by analogy with eq. 4) are jointly multivariate Gaussian.

Following Lee et al. (2017), to complete the argument, we need to show that the output feature maps are iid, i.e. $\mathbf{a}_i^{(\ell+1)}(\mathbf{X}, \mathbf{X}')$ and $\mathbf{a}_{i'}^{(\ell+1)}(\mathbf{X}, \mathbf{X}')$ are iid for $i \neq i'$. They are identically distributed, as $b_i^{(\ell+1)}$ and $\mathbf{W}_{i,j}^{(\ell+1)}$ are iid and $\phi(\mathbf{a}_j^{(\ell)}(\mathbf{X}, \mathbf{X}'))$ is shared. To show that they are independent, remember that $\mathbf{a}_i^{(\ell+1)}(\mathbf{X}, \mathbf{X}')$ and $\mathbf{a}_{i'}^{(\ell+1)}(\mathbf{X}, \mathbf{X}')$ are jointly Gaussian, so it is sufficient to show that they are uncorrelated, and we can show that they are uncorrelated because the weights, $\mathbf{W}_{i,j}^{(\ell+1)}$ are independent with zero-mean, eliminating any correlations that might arise through the shared RV, $\phi(\mathbf{a}_j^{(\ell)}(\mathbf{X}, \mathbf{X}'))$. In the appendix, we consider the more complex case where we take limits simultaneously.

## 3   THE CONVNET AND RESNET KERNELS

Here we derive a computationally efficient kernel corresponding to the CNN described in the previous section. It is surprising that we can compute the kernel efficiently because the feature maps,

$\mathbf{a}_i^{(\ell)}(\mathbf{X})$, display rich covariance structure due to the shared convolutional filter. Computing and representing these covariances would be prohibitively computationally expensive. However, in many cases we only need the variance of the output, e.g. in the case of classification or regression with a final dense layer. It turns out that this propagates backwards through the convolutional network, implying that for every layer, we only need the "diagonal covariance" of the activations: the covariance between the corresponding elements of $\mathbf{a}_j^{(\ell)}(\mathbf{X})$ and $\mathbf{a}_j^{(\ell)}(\mathbf{X}')$ (i.e. $\mathrm{diag}\left(\mathbb{C}\left[\mathbf{a}_j^{(\ell)}(\mathbf{X}), \mathbf{a}_j^{(\ell)}(\mathbf{X}')\right]\right)$).

## 3.1 GP MEAN AND COVARIANCE

A GP is completely specified by its mean and covariance (kernel) functions. These give the parameters of the joint Gaussian distribution of the RVs indexed by any two inputs, $\mathbf{X}$ and $\mathbf{X}'$. For the purposes of computing the mean and covariance, it is easiest to consider the network as being written entirely in index notation,

$$A_{i,\mu}^{(\ell+1)}(\mathbf{X}) = b_i^{(\ell+1)} + \sum_{j=1}^{C^{(\ell)}} \sum_{\nu=1}^{H^{(\ell)}D^{(\ell)}} W_{i,j,\mu,\nu}^{(\ell+1)} \phi(A_{j,\nu}^{(\ell)}(\mathbf{X})).$$

where $\ell$ and $\ell+1$ denote the input and output layers respectively, $j$ and $i \in \{1, \ldots, C^{(\ell+1)}\}$ denote the input and output channels, and $\nu$ and $\mu \in \{1, \ldots, H^{(\ell+1)}D^{(\ell+1)}\}$ denote the location within the input and output channel or feature-maps.

The mean function is thus easy to compute

$$\mathbb{E}\left[A_{i,\mu}^{(\ell+1)}(\mathbf{X})\right] = \mathbb{E}\left[b_i^{(\ell+1)}\right] + \sum_{j=1}^{C^{(\ell)}} \sum_{\nu=1}^{H^{(\ell)}D^{(\ell)}} \mathbb{E}\left[W_{i,j,\mu,\nu}^{(\ell+1)} \phi(A_{j,\nu}^{(\ell)}(\mathbf{X}))\right] = 0.$$

as $b_i^{(\ell+1)}$ and $W_{i,j,\mu,\nu}^{(\ell+1)}$ have zero mean, and $W_{i,j,\nu,\mu}^{(\ell+1)}$ are independent of the activations at the previous layer, $\phi(A_{j,\nu}^{(\ell)}(\mathbf{X}))$.

Now we show that it is possible to efficiently compute the covariance function. This is surprising because for many networks, we need to compute the covariance of activations between all pairs of locations in the feature map (i.e. $\mathbb{C}\left[A_{i,\mu}^{(\ell+1)}(\mathbf{X}), A_{i,\mu'}^{(\ell+1)}(\mathbf{X}')\right]$ for $\mu, \mu' \in \{1, \ldots, H^{(\ell+1)}D^{(\ell+1)}\}$) and this object is extremely high-dimensional, $N^2(H^{(\ell+1)}D^{(\ell+1)})^2$. However, it turns out that we only need to consider the "diagonal" covariance, (i.e. we only need $\mathbb{C}\left[A_{i,\mu}^{(\ell+1)}(\mathbf{X}), A_{i,\mu}^{(\ell+1)}(\mathbf{X}')\right]$ for $\mu \in \{1, \ldots, H^{(\ell+1)}D^{(\ell+1)}\}$), which is a more manageable quantity of size $N^2(H^{(\ell+1)}D^{(\ell+1)})$.

This is true at the output layer $(L+1)$: in order to achieve an output suitable for classification or regression, we use only a single output location $H^{(L+1)} = D^{(L+1)} = 1$, with a number of "channels" equal to the number of of outputs/classes, so it is only possible to compute the covariance at that single location. We now show that, if we only need the covariance at corresponding locations in the outputs, we only need the covariance at corresponding locations in the inputs, and this requirement propagates backwards through the network.

Formally, as the activations are composed of a sum of terms, their covariance is the sum of the covariances of all those underlying terms,

$$\mathbb{C}\left[A_{i,\mu}^{(\ell+1)}(\mathbf{X}), A_{i,\mu}^{(\ell+1)}(\mathbf{X}')\right] = \mathbb{V}\left[b_i^{(\ell)}\right] +$$
$$\sum_{j=1}^{C^{(\ell)}} \sum_{j'=1}^{C^{(\ell)}} \sum_{\nu=1}^{H^{(\ell)}D^{(\ell)}} \sum_{\nu'=1}^{H^{(\ell)}D^{(\ell)}} \mathbb{C}\left[W_{i,j,\mu,\nu}^{(\ell+1)} \phi(A_{j,\nu}^{(\ell)}(\mathbf{X})), W_{i,j',\mu,\nu'}^{(\ell+1)} \phi(A_{j',\nu'}^{(\ell)}(\mathbf{X}'))\right]. \tag{7}$$

As the terms in the covariance have mean zero, and as the weights and activations from the previous layer are independent,

$$\mathbb{C}\left[A_{i,\mu}^{(\ell+1)}(\mathbf{X}), A_{i,\mu}^{(\ell+1)}(\mathbf{X}')\right] = \sigma_b^2 +$$
$$\sum_{j=1}^{C^{(\ell)}} \sum_{j'=1}^{C^{(\ell)}} \sum_{\nu=1}^{H^{(\ell)}D^{(\ell)}} \sum_{\nu'=1}^{H^{(\ell)}D^{(\ell)}} \mathbb{E}\left[W_{i,j,\mu,\nu}^{(\ell+1)} W_{i,j',\mu,\nu'}^{(\ell+1)}\right] \mathbb{E}\left[\phi(A_{j,\nu}^{(\ell)}(\mathbf{X}))\phi(A_{j',\nu'}^{(\ell)}(\mathbf{X}'))\right]. \tag{8}$$

---

**Algorithm 1** The ConvNet kernel $k(\mathbf{X}, \mathbf{X}')$

---

1: *Input*: two images, $\mathbf{X}, \mathbf{X}' \in \mathbb{R}^{C^{(0)} \times (H^{(0)} W^{(0)})}$.
2: Compute $K_\mu^{(1)}(\mathbf{X}, \mathbf{X})$, $K_\mu^{(1)}(\mathbf{X}, \mathbf{X}')$, and $K_\mu^{(1)}(\mathbf{X}', \mathbf{X}')$ for $\mu \in \{1, \dots, H^{(1)} D^{(1)}\}$; using Eq. (10).
3: **for** $\ell = 1, 2, \dots, L$ **do**
4:     Compute $V_\mu^{(\ell)}(\mathbf{X}, \mathbf{X}')$, $V_\mu^{(\ell)}(\mathbf{X}, \mathbf{X}')$ and $V_\mu^{(\ell)}(\mathbf{X}, \mathbf{X}')$ for $\mu \in \{1, \dots, H^{(\ell)} D^{(\ell)}\}$; using Eq. (13), or some other nonlinearity.
5:     Compute $K_\mu^{(\ell+1)}(\mathbf{X}, \mathbf{X})$, $K_\mu^{(\ell+1)}(\mathbf{X}, \mathbf{X}')$, and $K_\mu^{(\ell+1)}(\mathbf{X}', \mathbf{X}')$ for $\mu \in \{1, \dots, H^{(\ell+1)} D^{(\ell+1)}\}$; using Eq. (11).
6: **end for**
7: Output the scalar $K_1^{(L+1)}(\mathbf{X}, \mathbf{X}')$.

---

The weights are independent for different channels: $\mathbf{W}_{i,j}^{(\ell+1)}$ and $\mathbf{W}_{i,j'}^{(\ell+1)}$ are iid for $j \neq j'$, so $\mathbb{E}\left[ W_{i,j,\mu,\nu}^{(\ell+1)} W_{i,j',\mu,\nu'}^{(\ell+1)} \right] = 0$ for $j \neq j'$ Further, each row $\mu$ of the weight matrices $\mathbf{W}_{i,j}^{(\ell+1)}$ only contains independent variables or zeros (Fig. 1), so $\mathbb{E}\left[ W_{i,j,\mu,\nu}^{(\ell+1)} W_{i,j',\mu,\nu'}^{(\ell+1)} \right] = 0$ for $\nu \neq \nu'$. Thus, we can eliminate the sums over $j'$ and $\nu'$:

$$\mathbb{C}\left[ A_{i,\mu}^{(\ell+1)}(\mathbf{X}), A_{i,\mu}^{(\ell+1)}(\mathbf{X}') \right] = \sigma_{\mathrm{b}}^2 + \sum_{j=1}^{C^{(\ell)}} \sum_{\nu=1}^{H^{(\ell)} D^{(\ell)}} \mathbb{E}\left[ W_{i,j,\mu,\nu}^{(\ell+1)} W_{i,j,\mu,\nu}^{(\ell+1)} \right] \mathbb{E}\left[ \phi(A_{j,\nu}^{(\ell)}(\mathbf{X})) \phi(A_{j,\nu}^{(\ell)}(\mathbf{X}')) \right]. \tag{9}$$

The $\mu$th row of $\mathbf{W}_{i,j}^{(\ell+1)}$ is zero for indices $\nu$ that do not belong to its convolutional patch, so we can restrict the sum over $\nu$ to that region. We also define $v_g^{(1)}(\mathbf{X}, \mathbf{X}')$, to emphasise that the covariances are independent of the output channel, $j$. The variance of the first layer is

$$K_\mu^{(1)}(\mathbf{X}, \mathbf{X}') = \mathbb{C}\left[ A_{i,\mu}^{(1)}(\mathbf{X}), A_{i,\mu}^{(1)}(\mathbf{X}') \right] = \sigma_{\mathrm{b}}^2 + \frac{\sigma_{\mathrm{w}}^2}{C^{(0)}} \sum_{i=1}^{C^{(0)}} \sum_{\nu \in \mu\text{th patch}} X_{i,\nu} X'_{i,\nu}. \tag{10}$$

And we do the same for the other layers,

$$K_\mu^{(\ell+1)}(\mathbf{X}, \mathbf{X}') = \mathbb{C}\left[ A_{i,\mu}^{(\ell+1)}(\mathbf{X}), A_{i,\mu}^{(\ell+1)}(\mathbf{X}') \right] = \sigma_{\mathrm{b}}^2 + \sigma_{\mathrm{w}}^2 \sum_{\nu \in \mu\text{th patch}} V_\nu^{(\ell)}(\mathbf{X}, \mathbf{X}'), \tag{11}$$

where

$$V_\nu^{(\ell)}(\mathbf{X}, \mathbf{X}') = \mathbb{E}\left[ \phi(A_{j,\nu}^{(\ell)}(\mathbf{X})) \phi(A_{j,\nu}^{(\ell)}(\mathbf{X}')) \right] \tag{12}$$

is the covariance of the activations, which is again independent of the channel.

### 3.2 COVARIANCE OF THE ACTIVITIES

The elementwise covariance in the right-hand side of Eq. (11) can be computed in closed form for many choices of $\phi$ if the activations are Gaussian. For each element of the activations, one needs to keep track of the 3 distinct entries of the bivariate covariance matrix between the inputs, $K_\mu^{(\ell+1)}(\mathbf{X}, \mathbf{X})$, $K_\mu^{(\ell+1)}(\mathbf{X}, \mathbf{X}')$ and $K_\mu^{(\ell+1)}(\mathbf{X}', \mathbf{X}')$.

For example, for the ReLU nonlinearity ($\phi(x) = \max(0, x)$), one can adapt Cho & Saul (2009) in the same way as Matthews et al. (2018a, section 3) to obtain

$$V_\nu^{(\ell)}(\mathbf{X}, \mathbf{X}') = \frac{\sqrt{K_\nu^{(\ell)}(\mathbf{X}, \mathbf{X}) K_\nu^{(\ell)}(\mathbf{X}', \mathbf{X}')}}{\pi} \left( \sin \theta_\nu^{(\ell)} + (\pi - \theta_\nu^{(\ell)}) \cos \theta_\nu^{(\ell)} \right) \tag{13}$$

where $\theta_\nu^{(\ell)} = \cos^{-1}\left( K_\nu^{(\ell)}(\mathbf{X}, \mathbf{X}') / \sqrt{K_\nu^{(\ell)}(\mathbf{X}, \mathbf{X}) K_\nu^{(\ell)}(\mathbf{X}', \mathbf{X}')} \right)$.

### 3.3 Efficiency of the ConvNet kernel

We now have all the pieces for computing the kernel, as written in Algorithm 1.

Putting together Eq. (11) and Eq. (13) gives us the surprising result that the diagonal covariances of the activations at layer $\ell + 1$ only depend on the diagonal covariances of the activations at layer $\ell$. This is very important, because it makes the computational cost of the kernel be within a constant factor of the cost of a forward pass for the equivalent CNN with 1 filter per layer.

Thus, the algorithm is more efficient that one would naively think. A priori, one needs to compute the covariance between all the elements of $\mathbf{a}_j^{(\ell)}(\mathbf{X})$ and $\mathbf{a}_j^{(\ell)}(\mathbf{X}')$ combined, yielding a $2H^{(\ell)}D^{(\ell)} \times 2H^{(\ell)}D^{(\ell)}$ covariance matrix for every pair of points. Instead, we only need to keep track of a $H^{(\ell)}D^{(\ell)}$-dimensional vector per layer and pair of points.

Furthermore, the particular form for the kernel (eq. 1 and eq. 2) implies that the required variances and covariances at all required locations can be computed efficiently as a convolution.

### 3.4 Kernel for a residual CNN

The induction step in the argument for GP behaviour from Sec. 2.2 depends only on the previous activations being iid Gaussian. Since all the activations are iid Gaussian, we can add skip connections between the activations of different layers while preserving GP behaviour, e.g. $\mathbf{A}^{(\ell+1)}$ and $\mathbf{A}^{(\ell-s)}$ where $s$ is the number of layers that the skip connection spans. If we change the NN recursion (Eq. 2) to

$$\mathbf{a}_i^{(\ell+1)}(\mathbf{X}) := \mathbf{a}_i^{(\ell-s)}(\mathbf{X}) + \mathbf{b}_i^{(\ell+1)} + \sum_{j=1}^{C^{(\ell)}} \mathbf{W}_{i,j}^{(\ell)} \phi\left(\mathbf{a}_j^{(\ell)}(\mathbf{X})\right), \tag{14}$$

then the kernel recursion (Eq. 11) becomes

$$K_\mu^{(\ell+1)}(\mathbf{X}, \mathbf{X}') = K_\mu^{(\ell-s)}(\mathbf{X}, \mathbf{X}') + \sigma_b^2 + \sigma_w^2 \sum_{\nu \in \mu\text{th patch}} V_\nu^{(\ell)}(\mathbf{X}, \mathbf{X}'). \tag{15}$$

This way of adding skip connections is equivalent to the "pre-activation" shortcuts described by He et al. (2016b). Remarkably, the natural way of adding residual connections to NNs is the one that performed best in their empirical evaluations.

## 4 Experiments

We evaluate our kernel on the MNIST handwritten digit classification task. Classification likelihoods are not conjugate for GPs, so we must make an approximation, and we follow Lee et al. (2017), in re-framing classification as multi-output regression.

The training set is split into $N = 50000$ training and 10000 validation examples. The regression targets $\mathbf{Y} \in \{-1, 1\}^{N \times 10}$ are a one-hot encoding of the example's class: $y_{n,c} = 1$ if the $n$th example belongs to class $c$, and $-1$ otherwise.

Training is exact conjugate likelihood GP regression with noiseless targets $\mathbf{Y}$ (Rasmussen & Williams, 2006). First we compute the $N \times N$ kernel matrix $\mathbf{K}_{xx}$, which contains the kernel between every pair of examples. Then we compute $\mathbf{K}_{xx}^{-1}\mathbf{Y}$ using a linear system solver.

The test set has $N_T = 10000$ examples. We compute the $N_T \times N$ matrix $\mathbf{K}_{x^*x}$, the kernel between each test example and all the training examples. The predictions are given by the row-wise maximum of $\mathbf{K}_{x^*x}\mathbf{K}_{xx}^{-1}\mathbf{Y}$.

For the "ConvNet GP" and "Residual CNN GP", (Table 1) we optimise the kernel hyperparameters by random search. We draw $M$ random hyperparameter samples, compute the resulting kernel's performance in the validation set, and pick the highest performing run. The kernel hyperparameters are: $\sigma_b^2$, $\sigma_w^2$; the number of layers; the convolution stride, filter sizes and edge behaviour; the nonlinearity (we consider the error function and ReLU); and the frequency of residual skip connections (for Residual CNN GPs). We do not retrain the model on the validation set after choosing hyperparameters.

| Method | #samples | Validation error | Test error |
|---|---|---|---|
| NNGP (Lee et al., 2017) | $\approx 250$ | – | 1.21% |
| Convolutional GP (van der Wilk et al., 2017) | SGD | – | 1.17% |
| Deep Conv. GP (Kumar et al., 2018) | SGD | – | 1.34% |
| ConvNet GP | 27 | 0.71% | 1.03% |
| Residual CNN GP | 27 | 0.72% | 0.96% |
| ResNet GP | – | 0.68% | **0.84%** |
| GP + parametric deep kernel (Bradshaw et al., 2017) | SGD | – | 0.60% |
| ResNet (Chen et al., 2018) | – | – | **0.41%** |

Table 1: MNIST classification results. #samples gives the number of kernels that were randomly sampled for the hyperparameter search. "ConvNet GP" and "Residual CNN GP" are random CNN architectures with a fixed filter size, whereas "ResNet GP" is a slight modification of the architecture by He et al. (2016b). Entries labelled "SGD" used stochastic gradient descent for tuning hyperparameters, by maximising the likelihood of the training set. The last two methods use parametric neural networks. The hyperparameters of the ResNet GP were not optimised (they were fixed based on the architecture from He et al., 2016b).

The "ResNet GP" (Table 1) is the kernel equivalent to a 32-layer version of the basic residual architecture by He et al. (2016a). The differences are: an initial $3 \times 3$ convolutional layer and a final dense layer instead of average pooling. We chose to remove the pooling because computing its output variance requires the off-diagonal elements of the filter covariance, in which case we could not exploit the efficiency gains described in Sec. 3.3.

We found that the despite it not being optimised, the 32-layer ResNet GP outperformed all other comparable architectures (Table 1), including the NNGP in Lee et al. (2017), which is state-of-the-art for non-convolutional networks, and convolutional GPs (van der Wilk et al., 2017; Kumar et al., 2018). That said, our results have not reached state-of-the-art for methods that incorporate a parametric neural network, such as a standard ResNet (Chen et al., 2018) and a Gaussian process with a deep neural network kernel Bradshaw et al. (2017).

To check whether the GP limit is applicable to relatively small networks used practically (with of the order of 100 channels in the first layers), we randomly sampled $10,000$ 32-layer ResNets, with 3, 10, 30 and 100 channels in the first layers, and, following the usual practice for ResNets we increase the number the number of hidden units when we downsample the feature maps. The probability density plots show a good match around 100 channels (Fig. 2A), which matches a more sensitive graphical procedure based on quantile-quantile plots (Fig. 2B). Notably, even for only 30 channels, the moments match closely (Fig. 2C). For comparison, typical ResNets use from 64 (He et al., 2016a) to 192 (Zagoruyko & Komodakis, 2016) channels in their first layers. We believe that this is because the moment propagation equations only require the Gaussianity assumption for propagation through the relu, and presumably this is robust to non-Gaussian input activations.

**Computational efficiency.** Asymptotically, computing the kernel matrix takes $O(N^2 LD)$ time, where $L$ is the number of layers in the network and $D$ is the dimensionality of the input, and inverting the kernel matrix takes $O(N^3)$. As such, we expect that for very large datasets, inverting the kernel matrix will dominate the computation time. However, on MNIST, $N^3$ is only around a factor of 10 larger than $N^2 LD$. In practice, we found that it was more expensive to *compute* the kernel matrix than to invert it. For the ResNet kernel, the most expensive, computing $\mathbf{K}_{xx}$, and $\mathbf{K}_{xx*}$ for validation and test took 3h 40min on two Tesla P100 GPUs. In contrast, inverting $\mathbf{K}_{xx}$ and computing validation and test performance took $43.25 \pm 8.8$ seconds on a single Tesla P100 GPU.

## 5 RELATED WORK

Van der Wilk et al. (van der Wilk et al., 2017) also adapted GPs to image classification. They defined a prior on functions $f$ that takes an image and outputs a scalar. First, draw a function $g \sim \mathcal{GP}(0, k_p(\mathbf{X}, \mathbf{X}'))$. Then, $f$ is the sum of the output of $g$ applied to each of the convolutional patches. Their approach is also inspired by convolutional NNs, but their kernel $k_p$ is applied to all pairs of patches of $\mathbf{X}$ and $\mathbf{X}'$. This makes their convolutional kernel expensive to evaluate, requiring

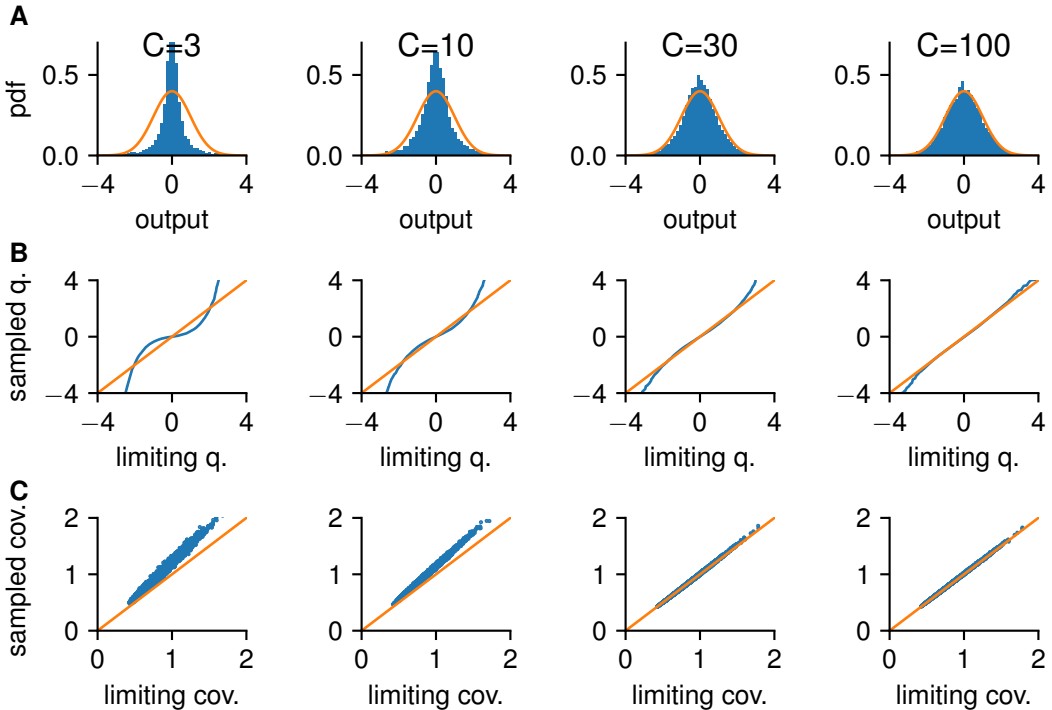

Figure 2: Comparison of the infinite limit, and outputs from finite 32-layer ResNets with 3, 10, 30, and 100 channels in their first layers. **A** Comparison of the empirical and limiting probability densities. **B** A more sensitive test of Gaussianity is a quantile-quantile plot, which shows converges with 100 channels. **C** The moments (variances and covariances) for 100 training inputs shows gives a good match for all numbers of channels.

inter-domain inducing point approximations to remain tractable. The kernels in this work, directly motivated by the infinite-filter limit of a CNN, only apply something like $k_p$ to the *corresponding* pairs of patches within $\mathbf{X}$ and $\mathbf{X}'$ (Eq. 10). As such, the CNN kernels are cheaper to compute and exhibit superior performance (Table 1), despite the use of an approximate likelihood function.

Kumar et al. (2018) define a prior over functions by stacking several GPs with van der Wilk's convolutional kernel, forming a "Deep GP" (Damianou & Lawrence, 2013). In contrast, the kernel in this paper confines all hierarchy to the definition of the kernel, and the resulting GPs is shallow.

Wilson et al. (2016) introduced and Bradshaw et al. (2017) improved deep kernel learning. The inputs to a classic GP kernel $k$ (e.g. RBF) are preprocessed by applying a feature extractor $g$ (a deep NN) prior to computing the kernel: $k_{\text{deep}}(\mathbf{X}, \mathbf{X}') := k(g(\mathbf{X}; \theta), g(\mathbf{X}', \theta))$. The NN parameters are optimised by gradient ascent using the likelihood as the objective, as in standard GP kernel learning (Rasmussen & Williams, 2006, Chapter 5). Since deep kernel learning incorporates a state-of-the-art NN with over $10^6$ parameters, we expect it to perform similarly to a NN applied directly to the task of image classification. At present both CNNs and deep kernel learning display superior performance to the GP kernels in this work. However, the kernels defined here have far fewer parameters (around 10, compared to their $10^6$).

Borovykh (2018) also suggests that a CNN exhibits GP behaviour. However, they take the infinite limit with respect to the *filter size*, not the number of filters. Thus, their infinite network is inapplicable to real data which is always of finite dimension.

Finally, there is a series of papers analysing the mean-field behaviour of deep NNs and CNNs which aims to find good random initializations, i.e. those that do not exhibit vanishing or exploding gradients or activations (Schoenholz et al., 2016; Yang & Schoenholz, 2017). Apart from their very different focus, the key difference to our work is that they compute the variance for a single training-

example, whereas to obtain the GPs kernel, we additionally need to compute the output covariances for different training/test examples (Xiao et al., 2018).

# 6    CONCLUSIONS AND FUTURE WORK

We have shown that deep Bayesian CNNs with infinitely many filters are equivalent to a GP with a recursive kernel. We also derived the kernel for the GP equivalent to a CNN, and showed that, in handwritten digit classification, it outperforms all previous GP approaches that do not incorporate a parametric NN into the kernel. Given that most state-of-the-art neural networks incorporate structure (convolutional or otherwise) into their architecture, the equivalence between CNNs and GPs is potentially of considerable practical relevance. In particular, we hope to apply GP CNNs in domains as widespread as adversarial examples, lifelong learning and k-shot learning, and we hope to improve them by developing efficient multi-layered inducing point approximation schemes.

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

# 7 APPENDIX

## 7.1 TECHNICAL NOTES ON LIMITS

The key technical issues in the proof (and the key differences between Lee et al. 2017 and Matthews et al. 2018b) arise from exactly how and where we take limits. In particular, consider the activations as being functions of the activities at the previous layer,

$$\mathbf{A}^{(4)} = \mathbf{A}^{(4)}(\mathbf{A}^{(3)}(\mathbf{A}^{(2)}(\mathbf{A}^{(1)}(\mathbf{X})))) \tag{16}$$

Now, there are two approaches to taking limits. First, both our argument in the main text, and the argument in Lee et al. (2017) is valid if we are able to take limits "inside" the network,

$$\mathbf{A}_{\mathrm{L}}^{(4)} = \lim_{C^{(3)} \to \infty} \mathbf{A}^{(4)} \left( \lim_{C^{(2)} \to \infty} \mathbf{A}^{(3)} \left( \lim_{C^{(1)} \to \infty} \mathbf{A}^{(2)} \left( \mathbf{A}^{(1)}(\mathbf{X}) \right) \right) \right). \tag{17}$$

However, Matthews et al. (2018a;b) argue that is preferable to take limits "outside" the network. In particular, Matthews et al. (2018b) take the limit with all layers simultaneously,

$$\mathbf{A}_{\mathrm{M}}^{(4)} = \lim_{n \to \infty} \mathbf{A}^{(4)} \left( \mathbf{A}^{(3)} \left( \mathbf{A}^{(2)} \left( \mathbf{A}^{(1)}(\mathbf{X}) \right) \right) \right), \tag{18}$$

where $C^{(\ell)} = C^{(\ell)}(n)$ goes to infinity as $n \to \infty$. That said, similar technical issues arise if we take limits in sequence, but outside the network.

## 7.2 EXTENDING THE DERIVATIONS OF MATTHEWS ET AL. (2018B) TO THE CONVOLUTIONAL CASE

In the main text, we follow Lee et al. (2017) in sequentially taking the limit of each layer to infinity (i.e. $C^{(1)} \to \infty$, then $C^{(2)} \to \infty$ etc.). This dramatically simplified the argument, because taking the number of units in the previous layer to infinity means that the inputs from that layer are exactly Gaussian distributed. However, Matthews et al. (2018b) argue that the more practically relevant limit is where we take all layers to infinity simultaneously. This raises considerable additional difficulties, because we must reason about convergence in the case where the previous layer is finite. Note that this section is not intended to stand independently: it is intended to be read alongside Matthews et al. (2018b), and we use several of their results without proof.

Mirroring Definition 3 in Matthews et al. (2018b), we begin by choosing a set of "width" functions, $C^{(\ell)}(n)$, for $\ell \in \{1, \ldots, L\}$ which all approach infinity as $n \to \infty$. In Matthews et al. (2018b), these functions described the number of hidden units in each layer, whereas here they describe the number of channels. Our goal is then to extend the proofs in Matthews et al. (2018b) (in particular,

of theorem 4), to show that the output of our convolutional networks converge in distribution to a Gaussian process as $n \to \infty$, with mean zero and covariance given by the recursion in Eqs. (10 – 12).

The proof in Matthews et al. (2018b) has three main steps. First, they use the Cramér-Wold device, to reduce the full problem to that of proving convergence of scalar random variables to a Gaussian with specified variance. Second, if the previous layers have finite numbers of channels, then the channels $\mathbf{a}_j^{(\ell)}(\mathbf{X})$ and $\mathbf{a}_j^{(\ell)}(\mathbf{X}')$ are uncorrelated but no longer independent, so we cannot apply the CLT directly, as we did in the main text. Instead, they write the activations as a sum of exchangeable random variables, and derive an adapted CLT for exchangeable (rather than independent) random variables (Blum et al., 1958). Third, they show that moment conditions required by their exchangeable CLT are satisfied.

To extend their proofs to the convolutional case, we begin by defining our networks in a form that is easier to manipulate and as close as possible to Eq. (21-23) in Matthews et al. (2018b),

$$A_{i,\mu}^{(1)} = f_{i,\mu}^{(1)}(x) = \frac{\sigma_{\mathrm{w}}}{\sqrt{C^{(0)}}} \sum_{j=1}^{C^{(0)}} \sum_{\nu \in \mu\text{th patch}} \epsilon_{i,j,\mu,\nu}^{(1)} x_{j,\nu} + b_i^{(1)}, \quad i \in \mathbb{N} \tag{19}$$

$$g_{i,\mu}^{(\ell)}(x) = \phi\left(f_{i,\mu}^{(\ell)}(x)\right) \tag{20}$$

$$A_{i,\mu}^{(\ell+1)} = f_{i,\mu}^{(\ell+1)}(x) = \frac{\sigma_{\mathrm{w}}}{\sqrt{C^{(\ell)}(n)}} \sum_{j=1}^{C^{(\ell)}(n)} \sum_{\nu \in \mu\text{th patch}} \epsilon_{i,j,\mu,\nu}^{(\ell+1)} g_{j,\nu}^{(\ell)}(x) + b_i^{(\ell+1)}, \quad i \in \mathbb{N} \tag{21}$$

where,

$$\epsilon_{i,j,\mu,\nu} \sim \mathcal{N}(0,1). \tag{22}$$

The first step is to use the Cramér-Wold device (Lemma 6 in Matthews et al., 2018b), which indicates that convergence in distribution of a sequence of finite-dimensional vectors is equivalent to convergence on all possible linear projections to the corresponding real-valued random variable. Mirroring Eq. 25 in Matthews et al. (2018b), we consider convergence of random vectors, $f_{i,\mu}^{(\ell)}(x)[n] - b_i^{(\ell)}$, projected onto $\alpha^{(x,i,\mu)}$,

$$\mathcal{T}^{(\ell)}(\mathcal{L}, \alpha)[n] = \sum_{(x,i,\mu) \in \mathcal{L}} \alpha^{(x,i,\mu)} \left[ f_{i,\mu}^{(\ell)}(x)[n] - b_i^{(\ell)} \right]. \tag{23}$$

where $\mathcal{L} \subset \mathcal{X} \times \mathbb{N} \times \{1, \ldots, H^{(\ell)} D^{(\ell)}\}$ is a finite set of tuples of data points and channel indicies, $i$, and indicies of elements within channels/feature maps, $\mu$. The suffix $[n]$ indicates width functions that are instantiated with input, $n$.

Now, we must prove that these projections converge in distribution a Gaussian. We begin by defining summands, as in Eq. 26 in Matthews et al. (2018b),

$$\gamma_j^{(\ell)}(\mathcal{L}, \alpha)[n] := \sigma_{\mathrm{w}} \sum_{(x,i,\mu) \in \mathcal{L}} \alpha^{(x,i,\mu)} \sum_{\nu \in \mu\text{th patch}} \epsilon_{i,j,\mu,\nu}^{(\ell)} g_{j,\nu}^{(\ell-1)}(x)[n], \tag{24}$$

such that the projections can be written as a sum of the summands, exactly as in Eq. 27 in Matthews et al. (2018b),

$$\mathcal{T}^{(\ell)}(\mathcal{L}, \alpha)[n] = \frac{1}{\sqrt{C^{(\ell-1)}(n)}} \sum_{j=1}^{C^{(\ell-1)}(n)} \gamma_j^{(\ell)}(\mathcal{L}, \alpha)[n]. \tag{25}$$

Now we can apply the exchangeable CLT to prove that $\mathcal{T}^{(\ell)}(\mathcal{L}, \alpha)[n]$ converges to the limiting Gaussian implied by the recursions in the main text. To apply the exchangeable CLT, the first step is to mirror Lemma 8 in Matthews et al. (2018b), in showing that for each fixed $n$ and $\ell \in \{2, \ldots, L + 1\}$, the summands, $\gamma_j^{(\ell)}(\mathcal{L}, \alpha)[n]$ are exchangeable with respect to the index $j$. In particular, we apply de Finetti's theorem, which states that a sequence of random variables is exchangeable if and

only if they are i.i.d. conditional on some set of random variables, so it is sufficient to exhibit such a set of random variables. Mirroring Eq. 29 in Matthews et al. (2018b), we apply the recursion,

$$\gamma_j^{(\ell)}(\mathcal{L}, \alpha)[n] := \sigma_{\mathrm{w}} \sum_{(x,i,\mu)\in\mathcal{L}} \alpha^{(x,i,\mu)} \sum_{\nu\in\mu\mathrm{th\ patch}} \epsilon_{i,j,\mu,\nu}^{(\ell)} \phi \left( \frac{\sigma_{\mathrm{w}}}{\sqrt{C^{(\ell-2)}(n)}} \sum_{k=1}^{C^{(\ell-2)}(n)} \sum_{\xi\in\nu\mathrm{th\ patch}} \epsilon_{j,k,\nu,\xi}^{(\ell-1)} g_{k,\xi}^{(\ell-2)}(x)[n] + b_j^{(\ell+1)} \right)$$
(26)

As such, the summands are iid conditional on the finite set of random variables $\left\{ g_{k,\xi}^{(\ell-2)}(x)[n] : k \in \{1,\ldots,C^{(\ell-2)}\}, \xi \in \{1,\ldots,H^{(\ell-2)}D^{(\ell-2)}\}, x \in \mathcal{L}_{\mathcal{X}} \right\}$, where $\mathcal{L}_{\mathcal{X}}$ is the set of input points in $\mathcal{L}$.

The exchangeable CLT in Lemma 10 in Matthews et al. (2018b) indicates that $\mathcal{T}^{(\ell)}(\mathcal{L}, \alpha)[n]$ converges in distribution to $\mathcal{N}\left(0, \sigma_*^2\right)$ if the summands are exchangeable (which we showed above), and if three conditions hold,

a) $\mathbb{E}_n\left[\gamma_j^{(\ell)}\gamma_{j'}^{(\ell)}\right] = 0$

b) $\lim_{n\to\infty} \mathbb{E}_n\left[\left(\gamma_j^{(\ell)}\right)^2 \left(\gamma_{j'}^{(\ell)}\right)^2\right] = \sigma_*^4$

c) $\mathbb{E}_n\left[|\gamma_j^{(\ell)}|^3\right] = o\left(\sqrt{C^{(\ell)}(n)}\right)$

Condition a) follows immediately as the summands are uncorrelated and zero-mean. Conditions b) and c) are more involved as convergence in distribution in the previous layers does not imply convergence in moments for our activation functions.

We begin by considering the extension of Lemma 20 in Matthews et al. (2018b), which allow us to show conditions b) and c) above, even in the case of unbounded but linearly enveloped nonlinearities (Definition 1 in Matthews et al., 2018b). Lemma 20 states that the eighth moments of $f_{i,\mu}^{(t)}(x)[n]$ are bounded by a finite constant independent of $n \in \mathbb{N}$. We prove this by induction. The base case is trivial, as $f_{j,\mu}^{(1)}(x)[n]$ is Gaussian. Following Matthews et al. (2018b), assume the condition holds up to $\ell - 1$, and show that the condition holds for layer $\ell$. Using Eq. (21), we can bound the activations at layer $\ell$,

$$\mathbb{E}\left[|f_{i,\mu}^{(\ell)}(x)[n]|^8\right] \leq 2^{8-1} \mathbb{E}\left[|b_i^{(\ell)}|^8 + \left|\frac{\sigma_{\mathrm{w}}}{\sqrt{C^{(\ell-1)}}} \sum_{j=1}^{C^{(\ell-1)}(n)} \sum_{\nu\in\mu\mathrm{th\ patch}} \epsilon_{i,j,\mu,\nu}^{(\ell)} g_{j,\nu}^{(\ell-1)}(x)[n]\right|^8\right]$$
(27)

Following Eq. 48 in Matthews et al. (2018b), which uses Lemma 19 in Matthews et al. (2018b), we have,

$$\mathbb{E}\left[\left|\frac{\sigma_{\mathrm{w}}}{\sqrt{C^{(\ell-1)}}} \sum_{j=1}^{C^{(\ell-1)}(n)} \sum_{\nu\in\mu\mathrm{th\ patch}} \epsilon_{i,j,\mu,\nu}^{(\ell)} g_{j,\nu}^{(\ell-1)}(x)[n]\right|^8\right]$$
$$= \frac{2^4\Gamma(4+1/2)}{\Gamma(1/2)} \mathbb{E}\left[\left|\frac{\sigma_{\mathrm{w}}^2}{C^{(\ell-1)}(n)} \|g_{j\in\{1,\ldots,C^{(\ell-1)}(n)\},\nu\in\mu\mathrm{th\ patch}}^{(\ell-1)}(x)[n]\|_2^2\right|^4\right].$$
(28)

where $g_{j\in\{1,\ldots,C^{(\ell-1)}(n)\},\nu\in\mu\mathrm{th\ patch}}^{(\ell-1)}(x)[n]$ is the set of post-nonlinearities corresponding to $j \in \{1,\ldots,C^{(\ell-1)}(n)\}$ and $\nu \in \mu$th patch. Following Matthews et al. (2018b), observe that,

$$\frac{1}{C^{(\ell-1)}(n)} \|g_{j\in\{1,\ldots,C^{(\ell-1)}(n)\},\nu\in\mu\mathrm{th\ patch}}^{(\ell-1)}(x)[n]\|_2^2 = \frac{1}{C^{(\ell-1)}(n)} \sum_{j=1}^{C^{(\ell-1)}(n)} \sum_{\nu\in\mu\mathrm{th\ patch}} \left(g_{j,\nu}^{(\ell-1)}(x)[n]\right)^2$$
(29)

$$\leq \frac{1}{C^{(\ell-1)}(n)} \sum_{j=1}^{C^{(\ell-1)}(n)} \sum_{\nu\in\mu\mathrm{th\ patch}} \left(c + m|f_{j,\nu}^{(\ell-1)}(x)[n]|\right)^2$$
(30)

by the linear envelope property, $|\phi(u)| \leq c + m|u|$. Following Matthews et al. (2018b), we substitute this bound back into Eq. (28) and suppress a multiplicative constant independent of $x$ and $n$,

$$\mathbb{E}\left[\left|\frac{\sigma_{\mathrm{w}}}{\sqrt{C^{(\ell-1)}(n)}} \sum_{j=1}^{C^{(\ell-1)}(n)} \sum_{\nu \in \mu\text{th patch}} \epsilon_{i,j,\mu,\nu}^{(\ell)} g_{j,\nu}^{(\ell-1)}(x)[n]\right|^8\right]$$

$$\leq \frac{1}{\left(C^{(\ell-1)}(n)\right)^4} \mathbb{E}\left[\left|\sum_{j=1}^{C^{(\ell-1)}(n)} \sum_{\nu \in \mu\text{th patch}} c^2 + 2cm|f_{j,\mu}^{(\ell-1)}(x)[n]| + m^2|f_{j,\mu}^{(\ell-1)}(x)[n]|^2\right|^4\right] \quad (31)$$

This can be multiplied out, yielding a weighted sum of expectations of the form,

$$\mathbb{E}\left[|f_{k,\nu}^{(\ell-1)}(x)[n]|^{p_1}|f_{l,\xi}^{(\ell-1)}(x)[n]|^{p_2}|f_{r,\pi}^{(\ell-1)}(x)[n]|^{p_3}|f_{q,\rho}^{(\ell-1)}(x)[n]|^{p_4}\right] \quad (32)$$

with $p_i \in \{0, 1, 2\}$ for $i = 1, 2, 3, 4$, and $k, l, r, q \in \{1, \ldots, C^{(\ell-1)}(n)\}$, and $\nu, \xi, \pi, \rho \in \mu$th patch where the weights of these terms are independent of $n$. Using Lemma 18 in Matthews et al. (2018b), each of these terms is bounded if the eighth moments of $f_{k,\mu}^{(\ell-1)}(x)[n]$ are bounded, which is our inductive hypothesis. The number of terms in the expanded sum is upper bounded by $\left(2C^{(\ell-1)}(n)|\mu\text{th patch}|\right)^4$, where $|\mu$th patch$|$ is the number of elements in a convolutional patch. Thus, we can use the same constant for any $n$ due to the $1/\left(C^{(\ell-1)}(n)\right)^4$ scaling. As in Matthews et al. (2018b), noting that $f_{j,\mu}^{(\ell-1)}(x)[n]$ are exchangeable over $j$ for any $x$ and $n$ concludes the proof.

Using this result, we can obtain a straightforward adaptation of Lemmas 15, 16 and 21 in Matthews et al. (2018b). Lemma 15 gives condition b), Lemma 16 gives condition c); Lemma 15 requires uniform integrability, which is established by Lemma 21.

### 7.3 CALIBRATION OF GAUSSIAN PROCESS UNCERTAINTY

It is important to check that the estimates of uncertainty produced by our Gaussian process are reasonable. However, to make this assessment, we needed to use a proper likelihood, and not the squared-error loss in the main text. We therefore used our kernel to perform the full, multi-class classification problem in GPflow Matthews et al. (2017), with a RobustMax likelihood (Hernández-lobato et al., 2011). The more difficult non-conjugate inference problem forced us to use 1000 inducing points, randomly chosen from the training inputs. Both our kernel and an RBF kernel have similar calibration curves, that closely track the diagonal, indicating accurate uncertainty estimation. However, even in the inducing point setting, our convolutional kernel gave considerably better performance than the RBF kernel (2.4% error vs 3.4% error).

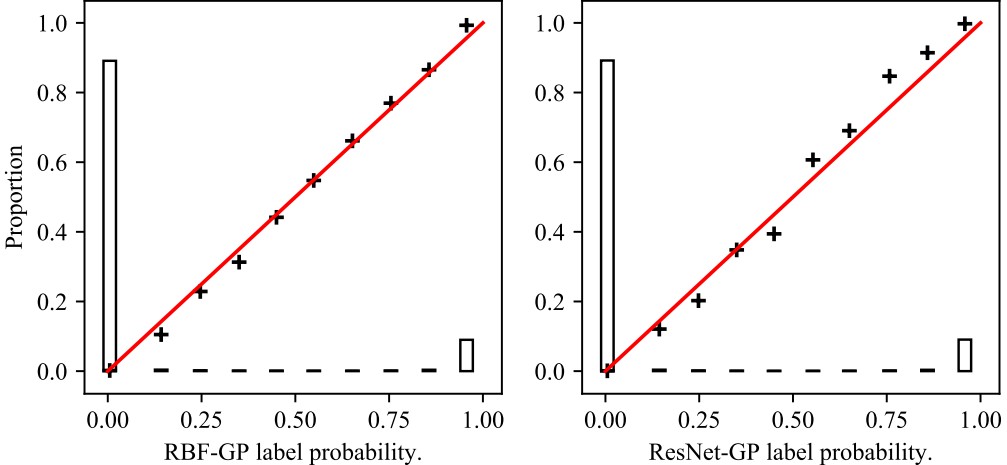

Figure 3: Calibration plots for an RBF kernel (left) and the ResNet kernel (right). The x-axis gives GP prediction for the label probability. The points give corresponding proportion of test points with that label, and the bars give the proportion of training examples in each bin.

