# OpenReview forum: "Deep Convolutional Networks as shallow Gaussian Processes"
_ICLR.cc/2019/Conference_

### Official Review · AnonReviewer1 · 2018-11-02
**An efficient convolutional kernel for GPs without proper evaluation of probabilistic predictions**

**Rating:** 5
**Confidence:** 4

**Review:**

This paper shows that deep convolutional networks (CNNs, without pooling) with a suitable prior over weights can be seen as shallow Gaussian processes (GPs) with a specific covariance function. It shows that this covariance function can be computed efficiently (when compared to previous attempts at resembling convolutional networks with GPs), with a cost that only depends linearly on the number of layers and the input dimensionality, i.e.~O(N^2 L D).

To show the equivalence between deep CNNs and shallow GPs, the paper uses similar ideas to those proposed by Matthews et al (2018a) and Lee et al (2017), i.e. using the multivariate central limit theorem in very large networks, where in the case of this paper the limit is taken as the number of channels at each layer goes to infinity. Therefore, from a theoretical perspective, these ideas have been proposed before. However, the paper presents a novel efficient way to compute the convolutional kernel, which I believe has merits on its own.

However, the model setting for classification (where deep CNNs have been successful) and consequent evaluation on the MNIST dataset is less than convincing. One of the main motivations for Bayesian CNNs and GPs (and the paper argue for this in the intro) is to be able to provide good uncertainty estimates. However, the classification problem is framed in a regression setting, where neither probabilistic estimates are evaluated or even provided. Indeed, only the error rate is given on Table 1. To me, this is certainly not enough for a Bayesian/GP method and it is a critical deficiency of the paper in its current form. While I understand having a non-Gaussian likelihood will complicate things and conflate the kernel contribution with the approximations, I believe it is necessary to provide and evaluate such probabilistic estimates and compare them to other GP approaches (even using other less than satisfying methods such as calibration/scaling). Along a similar vein, it is unclear what objective function was used for hyper-parameter learning but, given that the authors actually “sample hyper-parameters”, I am guessing a proper probabilistic objective such as the marginal likelihood is out of the question.

Other (perhaps minor) deficiencies is that the method is not scalable to large datasets (I am even surprised the authors managed to run this on full MNIST) and that no theoretical analysis is done (e.g. as in Mattews et al, 2018a).

Minor comments:

* In the intro, “Other methods such as Gaussian Processes”: GPs are not a method and I believe the authors really mean here Gaussian process regression.
* The prior variance over filters in Eq (3) divides over the number of channels.  Why does a Gaussian prior with infinite precision make sense here?
* The authors should report the state of the art of using GPs for MNIST classification using non-convolutional kernels).

---

> ### Author Response · Authors · 2018-11-26
> **Response concerning uncertainty estimates, scalability and theoretical analysis.**
>
> Thank you very much for your comments.
>
> To check the quality of the uncertainties produced by our method, we used GPflow to perform the multi-class classification problem, on the full dataset, with a RobustMax likelihood. To ensure tractability in the harder case with a non-conjugate likelihood, we were forced to use sparse variational inference with 1,000 inducing points chosen randomly from the training set.  (Though we expect that carefully optimized code exploiting the GPU for matrix inverses could solve the full problem in the non-conjugate setting).  The resulting calibration curves were close to the diagonal, indicating accurate uncertainty estimation.  We compared to a GP with an RBF kernel in the same setting, which gave a similarly good calibration curve, but worse test error (3.4% vs 2.4%).
>
> As regards hyper-parameter optimization, it is important to note that our best architecture, the ResNet GP, was not optimized *at all*, we simply used the 32-layer ResNet architecture directly, replacing the final average-pooling layer with a dense layer.  For the lower-performing architectures (ConvNet GP and Residual CNN GP), we followed Lee et al. (2017) in randomly sampling hyperparameters including e.g. the number of layers, and selected the network with the best validation error.
>
> Regarding scalability (as we discuss above in our response to reviewer 1), inverting the kernel matrix takes around a minute (though this can be reduced further by using MAGMA with multiple GPUs), whereas computing the kernel matrix takes longer.  Critically, however, computing the kernel matrix is embarassingly parallel, so can be speeded up arbitrarily given sufficiently many GPU's.  In contrast, neural network training is an inherently sequential computation, and as such improving training speed with additional GPU's remains an active research topic.  Considering the question of practical applicability more broadly, it is important to make two points.  First, we agree that there is a need for approximations schemes that are effective in the large-scale neural network domain, and we look forward to future research on this topic.  Second, the Gaussian process solves a *really* hard problem: exact inference over all the parameters in an infinitely wide, multilayer ResNet.  It is pretty shocking (to us at least) that this problem is tractable at all.  In contrast, doing full-gradient HMC for a finite version of this network would be practically impossible.
>
> We have added a new section in the Appendix, which describes how to extend the results in Matthews et al 2018b to our case.
>
> Minor comments:
> * Thanks, fixed.
> * If we didn't decrease the weights as we increased the number of channels, then the neural network outputs would blow up as we increased the number of channels, because the outputs would be the sum of an increasingly large number of terms.  We decrease the weights as we increase the number of channels (as in Lee et al. 2017, and Matthews et al. 2018) to ensure that the neural network outputs have a sensible scale, even as we take the number of channels to infinity.
> * Lee et al. 2017 used a non-convolutional kernel, which is state-of-the-art for nonconvolutional networks, and we have noted this in the text.

---

### Official Review · AnonReviewer3 · 2018-11-03
**GP kernel inspired by CNNs outperforms previous non-parametric approaches**

**Rating:** 8
**Confidence:** 3

**Review:**

This paper

1) extends an argument for the GP behaviour of deep, infinitely-wide fully-connected networks to convolutional and residual deep neural networks with infinitely many channels and
2) provides a computationally tractable approach to compute the corresponding GP kernel. This kernel has few hyper-parameters, and achieves state-of-the-art results on the MNIST dataset.

While point (1) is a relatively straightforward adaptation of Lee et. al (2017) and Matthews et al. (2018) to a different network structure, point (2) is original and non-trivial. All in all, I think this paper makes a significant contribution that I believe will spark interesting follow-up work (hinted at in the last section of the paper).

Questions:

- In my understanding, the kernels of Section 3 do not require the weight matrices W to share the same values across rows. Accordingly, their performance cannot necessarily be explained by properties of convolutional filters (in particular translation invariance). Can the authors comment on that?
- What would be the performance of a parametric CNN trained with SGD that matches the architecture (# layers) & the squared loss function of ResNet GP? The only point of comparison is Chen & al. (2018), which I suppose optimizes a log loss? Specifically, I would like to understand the impact of the loss function and of the number of layers on the relative performance of the two approaches.

The paper is clear and easy to follow. A few suggestions:

- I recommend turning the argument in section 2.2 into a formal, self-contained theorem that states a result on A_L, defined in eq. 17 (which I would move to the main text). This would make the precise claim easier to understand.
- I suggest including a more thorough discussion of the results. Table 1 is only introduced in the related work section.
- If space is a concern, I would move part of Section 2.2 outside of the main text, since it mostly follows Lee et al. & Matthews et al

Small questions/comments:

- Eqs 1 and 2: b_j should be multiplied by the all-ones vector, just like in (5) and (6).
- Below eq. 5: "while the *elements of the* feature maps themselves display..."
- Paragraph above eq. 7: "in order to achieve an output suitable for *binary* classification or *univariate* regression"
- Paragraph above eq. 7: "if we only need the covariance at *certain* locations in the outputs..."
- Algorithm 1: you might want to add a loop over g for clarity

---

> ### Author Response · Authors · 2018-11-26
> **Response discussing weight tying, and finite vs infinite CNNs**
>
> Thank you very much for your comments.
>
> Questions
> - You're right that we design our networks in such a way that the weight-sharing in the convolutional filters becomes irrelevant.  However, it is important to note that this only occurs in restricted circumstances.  In particular, weight-sharing introduces correlations across different locations in a single feature map.  It turns out that in many cases, you can design neural network priors with kernels that don't depend on these correlations, and that's what we do.  However, for many neural networks, the correlations (and hence the weight sharing) will be relevant.  For instance, if we use any type of pooling (such as average pooling at the output), then we would require the correlations within the feature maps, and hence we would need to take into account covariances and hence weight sharing throughout the whole network.
> - We agree that a comparison of the performance of finite and infinite CNNs as regards the loss-function, and the number of layers is important.  Unfortunately, we weren't able to satisfy ourselves that we had a thorough, completely fair comparison in the available time, given the many choices available for SGD, so we'll restrict ourselves to three observations here.  First, using proper likelihoods rather than the squared loss function in the GP appears to give little or no improvement in test error, though this may be because we need additional approximations. Second, depth is critical to achieving higher performance, with shallower architectures such as our ConvNet GP and Residual CNN GP (with a maximum of 16 layers) achieving a test error of around 1.0%, as opposed to 0.84% with a deeper 32 layer ResNet.  Third, while our architecture is very similar to a ResNet (the only real difference being the replacement of average pooling at the last layer with a dense layer), the performance does not equal that of state-of-the-art residual networks. A thorough investigation of this phenomenon will form an important avenue for future research.
>
> Suggestions:
> We have given a more thorough discussion of the results, including by introducing Table 1 in the Experiments section.  We've also added a section to the appendix which extends the proofs in Matthews et al. (2018b) to the convolutional case.
>
> Small questions/comments:
> - Thanks: fixed.
> - Thanks: fixed.
> - Even for multivariate regression, this is true: the output channels, not locations correspond to different classes.  We have clarified this in the text.  It would also be interesting to think about image-scale outputs, which would correspond to e.g. classifying each pixel/region in an image.
> - We need the covariances across data at all corresponding locations in the in the feature map.  We have clarified this formally in the text.
> - We have explicitly noted that the kernel is computed for all mu (previously g).

---

### Official Review · AnonReviewer2 · 2018-11-06
**nice direction of research, but limited applicability**

**Rating:** 5
**Confidence:** 5

**Review:**

The current paper considers the relation between convolutional neural networks and Gaussian processes from theoretical and practical point of view.

The main contribution of the paper as presented by the authors is 2-fold:
1. Some theoretical justifications about the correspondence between GPs and convolutional networks with infinitely many channels are provided.
2. The formulas for GP kernel computation for the considered network is provided and some experiments are conducted (on MNIST).

I personally enjoy the ideas of the close relation between certain types of neural networks and GPs and I really like the idea of authors that kernels based on convolutional networks might be more practical compared to the ones based on fully connected networks. It might be easier to encode certain invariance for complex objects via multilayered structures then via more simple explicit kernels.

However, I see couple of important issues:
1. The theoretical justification provided is basically heuristic argument and, speaking rigorously, is not a theorem. The proper proof should be based not on layer-by-layer convergence, but on the convergence with all the parameters tending simultaneously to infinity (see Matthews et al, 2018). Also, I doubt that the limit with infinite number of channels is as meaningful as the limit with infinite width of layer, as wide networks are used much more often in practice than networks with many channels.
2. The practical applicability is very limited as the kernel obtained has very high computational complexity. The authors theirselves comment that computing kernel matrix takes more time than inverting it. Thus, the applicability beyond MNIST is a big question for the proposed approach.

To sum up, I think that the present paper targets an important direction of work, but the contribution itself is somehow limited (and relative obvious based on the recent papers on relation between fully connected networks and GPs).

---

> ### Author Response · Authors · 2018-11-26
> **Response concerning extending Matthews et al. 2018, empirical comparison of finite and infinite priors, and scalability.**
>
> Thank you very much for your comments.
>
> 1.) We have added a section in the Appendix, where we extend the proof by Matthews et al. (2018) to the convolutional case.
>
> We have also done a new experiment looking at the behaviour of randomly sampled, finite convolutional networks (Fig. 2).  This experiment shows that for 100 filters in the 1st convolutional layers, the behaviour of finite networks closely matches the infinite limit: the marginals are close to Gaussian, and the moments match closely. For comparison, typical ResNets use from 64 (He et al, 2016a) to 192 (Zagoruyko & Komodakis, 2016) channels in their first layers.
>
> 2.) Indeed, inverting the kernel matrix takes around a minute (though this can be reduced further by using MAGMA with multiple GPUs), whereas computing the kernel matrix takes longer.  Critically, however, computing the kernel matrix is embarassingly parallel, so can be speeded up arbitrarily given sufficiently many GPUs.  In contrast, neural network training is inherently sequential, and as such improving training speed with additional GPUs remains an active research topic. In fact, one of our key results is to show that computing the kernel is surprisingly tractable, as special properties of the kernel corresponding to CNNs can be exploited such that the kernel computation for a pair of images becomes equivalent to a pass through a single-channel CNN.
>
> Considering the question of practical applicability more broadly, it is important to make two points. First, we agree that there is a need for approximations schemes that are effective in the large-scale neural network domain, and we look forward to future research on this topic. Second, the Gaussian process solves a *really* hard problem: exact inference over all the parameters in an infinitely wide, multilayer ResNet.  It is surprising that this problem is tractable at all.  In contrast, doing full-gradient HMC for a finite version of the 32-layer ResNet is unlikely to be tractable.

---

### Meta-Review · Area_Chair1 · 2018-12-13
**A neat connection between deep convolutional networks and Gaussian processes**

**Confidence:** 5
**Recommendation:** Accept (Poster)

**Metareview:**

This paper builds on a promising line of literature developing connections between Gaussian processes and deep neural networks.  Viewing one model under the lens of (the infinite limit of) another can lead to neat new insights and algorithms.  In this case the authors develop a connection between convolutional networks and Gaussian processes with a particular kind of kernel.  The reviews were quite mixed with one champion and two just below borderline.

The reviewers all believed the paper had contributions which would be interesting to the community (such as R1: "the paper presents a novel efficient way to compute the convolutional kernel, which I believe has merits on its own" and R2: "I really like the idea of authors that kernels based on convolutional networks might be more practical compared to the ones based on fully connected networks").  All the reviewers found the contribution of the covariance function to be novel and exciting.

Some cited weaknesses of the paper were that the authors didn't analyze the uncertainty from the model (arguably the reasoning for adopting a Bayesian treatment), novelty in appealing to the central limit theorem to arrive at the connection, and scalability of the model.

In the review process it also became apparent that there was another paper with a substantially similar contribution.  The decision for this paper was calibrated accordingly with that work.

Weighing the strengths and weaknesses of the paper and taking into account a reviewer willing to champion the work it seems there is enough novel contribution and interest in the work to justify acceptance.

The authors provided responses to the reviewer concerns including calibration plots and timing experiments in the discussion period and it would be appreciated if these can be incorporated into the camera ready version.